# A Novel Reinforcement Learning Collision Avoidance Algorithm for USVs Based on Maneuvering Characteristics and COLREGs

**DOI:** 10.3390/s22062099

**Published:** 2022-03-08

**Authors:** Yunsheng Fan, Zhe Sun, Guofeng Wang

**Affiliations:** 1College of Marine Electrical Engineering, Dalian Maritime University, Dalian 116026, China; rickey628@dlmu.edu.cn (Z.S.); dmuwgf@dlmu.edu.cn (G.W.); 2Key Laboratory of Technology and System for Intelligent Ships of Liaoning Province, Dalian 116026, China

**Keywords:** unmanned surface vehicle, deep reinforcement learning, autonomous collision avoidance, COLREGs

## Abstract

Autonomous collision avoidance technology provides an intelligent method for unmanned surface vehicles’ (USVs) safe and efficient navigation. In this paper, the USV collision avoidance problem under the constraint of the international regulations for preventing collisions at sea (COLREGs) was studied. Here, a reinforcement learning collision avoidance (RLCA) algorithm is proposed that complies with USV maneuverability. Notably, the reinforcement learning agent does not require any prior knowledge about USV collision avoidance from humans to learn collision avoidance motions well. The double-DQN method was used to reduce the overestimation of the action-value function. A dueling network architecture was adopted to clearly distinguish the difference between a great state and an excellent action. Aiming at the problem of agent exploration, a method based on the characteristics of USV collision avoidance, the category-based exploration method, can improve the exploration ability of the USV. Because a large number of turning behaviors in the early steps may affect the training, a method to discard some of the transitions was designed, which can improve the effectiveness of the algorithm. A finite Markov decision process (MDP) that conforms to the USVs’ maneuverability and COLREGs was used for the agent training. The RLCA algorithm was tested in a marine simulation environment in many different USV encounters, which showed a higher average reward. The RLCA algorithm bridged the divide between USV navigation status information and collision avoidance behavior, resulting in successfully planning a safe and economical path to the terminal.

## 1. Introduction

The application of artificial intelligence algorithms to navigation is a staple of research in USVs’ control. With the increasing requirements for the autonomous and intelligent level of USVs, artificial intelligence naturally has been given more attention. Although the concept of reinforcement learning was proposed earlier, it was only when the success of Google DeepMind’s application of deep reinforcement learning in Atari [1,2], StarCraft II [3], and Go [4] that it attracted worldwide attention. In these projects, the control ability of the agent can reach the human level or even be better than humans in some aspects. The reinforcement learning algorithm has achieved many results in a variety of domains. There has been much innovative research in unmanned system, especially the autonomous navigation system of USVs [5]. There are many issues to consider in the design of the collision avoidance algorithm:(1)The COLREGs need to be followed, which divide the responsibilities and restrict the behaviors when encountering other USVs;(2)The collision avoidance algorithm should be able to work stably and reliably for a long time;(3)The algorithm for multi-USV collision avoidance is critical;(4)The algorithm should have a tremendous real-time performance to face the real-time changes of the marine environment.

Much progress towards USV collision avoidance has been made using reinforcement learning algorithms that optimize their control of an environment well.

Some researchers focused on elaborate reward design. Cheng and Zhang [6] proposed the reward functions for under-actuated USVs with full consideration of interference and ship maneuverability. The trained USV agent planned the path in a simulation marine environment with a large number of static obstacles existing successfully. Chun and Roh [7] proposed a new method for calculating the collision risk of ships using the ship domain and the closest point of approach (CPA). A fine collision avoidance path that follows the COLREGs was generated by the reinforcement learning proximal policy optimization (PPO) algorithm [8]. This algorithm is more effective than the traditional A* algorithm. Li and Wu [9] improved the DQN algorithm reward function with the artificial potential field algorithm (APF) according to the ship maneuverability and COLREGs.

Some researchers focused on algorithm innovation. Xie and Chu [10] proposed the collision avoidance algorithm based on the asynchronous advantage actor–critic (A3C) [11] in the on-policy deep reinforcement learning algorithm. This asynchronous algorithm can make up for the shortcomings of the slow learning speed and non-convergence of the on-policy actor–critic algorithm. To improve the USV collision avoidance effect, a long short-term memory neural network (LSTM) was used. Wu and Chen [12] optimized the performance of the deep Q network based on the dueling network architecture. The improved algorithm obtained higher exploration efficiency and convergence speed than DQN and Deep SARSA [13,14]. Guo and Zhang [15] designed the collision avoidance algorithm based on the deep deterministic policy gradient (DDPG) algorithm [16], which can solve the problem of the difficulty in the convergence of the actor–critic (AC) method. The collision avoidance agent is trained by real-time ship information provided by the ship AIS system. In the simulation of the electronic chart, the convergence was faster than the DQN and AC algorithms, and the average reward was higher.

Some researchers focused on the construction of a characteristic training environment. Woo and Kim [17] designed a ship collision avoidance algorithm by the deep reinforcement learning algorithm. It was combined with the semi-Markov decision process, which designed the action as a complex process such as path following and starboard avoidance. Using the velocity obstacle algorithm for vessels’ motion control, a good collision avoidance effect was obtained in a grid map. Xu and Lu [18] took the control force or yaw moment under the Fossen ship model as the input [19]. The collision avoidance effect was better than the artificial potential field algorithm and the velocity obstacle algorithm under the unity simulation environment. Zhao and Roh [20] divided the ship surrounding marine environment into four regions under the unstructured and unpredictable marine environment, and the complex states of the ship were mapped to the rudder angle control command. Considering the nearest obstacle ship, the multi-ship collision avoidance algorithm was designed based on DQN. Zhang and Wang [21] divided the collision avoidance marine environment into the scene division layer and the autonomous navigation decision-making layer. To solve the existing problems of falling into the local optimum and the slow convergence speed of the DQN algorithm, they introduced APF for the terminal, which effectively solved the problem and increased the convergence speed. Shen and Hashimoto [22] evaluated the collision risk by the bumper ship domain model. A deep reinforcement learning algorithm for USV collision avoidance with the dueling network architecture was designed. The collision avoidance algorithm was experimented with in a pool and successfully avoided the obstacle boats. Zhou and Wu [23] applied the deep reinforcement learning algorithm to the formation path planning. The USVs avoided obstacles successfully. The formation of ships did not collide with each other.

Model-free reinforcement learning has certain advantages in the design of autonomous collision avoidance algorithms for unmanned vehicles. Firstly, it is difficult to accurately model the situation of the USVs, but the reinforcement learning algorithm can solve this problem well by identifying the process in the control. Secondly, compared with other intelligent collision avoidance algorithms, the deep reinforcement learning algorithm does not rely on labeled samples to solve the problem of collision avoidance. Finally, compared with other traditional collision avoidance algorithms, the method of learning is more consistent with the way humans solve problems.

In this paper, an autonomous collision avoidance algorithm, RLCA, is proposed. The main contributions of this paper are as follows:(1)A characteristic Markov decision process for USV collision avoidance that fully considers the COLREGs and maneuverability of USVs was built. Remarkably, the agent realizes the training process without learning the coordinate information of the longitude and latitude. This makes the algorithm more realistic;(2)The double-DQN method was used to reduce the overestimation in the training process. The architecture of the deep Q network was improved by the dueling architecture; it can describe the value function in more detail. Furthermore, the way of storing training transitions was improved;(3)To solve the exploration problem of the agent, an optimized exploration method called category-based exploration suitable for the characteristics of the USV is proposed, which can effectively improve the exploration ability of the USV agents;(4)The RLCA algorithm can achieve an outstanding collision avoidance effect in many typical encounter situations. Specifically, the RLCA agent can obtain a higher average reward in training and has good engineering prospects.

This paper is organized as follows. The COLREGs and mathematical models of USVs are given in Section 2. The details of the autonomous collision avoidance algorithm with reinforcement learning for USVs are given in Section 3. Section 4 gives the description of the RLCA algorithm. In Section 5, the training and simulation of the RLCA algorithm with single or multiple obstacle USVs are carried out. The last section is the summary and prospects.

## 2. Mathematical Model and COLREGs

The autonomous collision avoidance algorithm should conform to the maneuverability and COLREGs of USVs. The key principle is that an excellent collision avoidance behavior enables avoiding obstacle USVs safely and planning a path to the terminal economically.

### 2.1. The COLREGs for USVs

That the COLREGs can constrain the USV behavior in encounters well was a significant basis for the algorithm design. It needs to judge whether the own USV needs to carry out collision avoidance motions [24] when the own USV ($USV_{U}$) senses the information of the obstacle USV ($USV_{O}$). If it is considered that the collision avoidance process is over, the original navigation state should be resumed immediately. According to the requirements of the COLREGs, the encounter cases of USVs are divided into three situations at different angles in Figure 1. As shown in Figure 2, according to the situation division in [18], for different encounters, different collision avoidance behaviors are performed according to the COLREGs.

Figure 2a shows the head-on USV encounter situation, in which the two USVs are navigating in opposite or close to opposite routes with a danger of collision. Both USVs have a collision avoidance duty to turn to the starboard side. Figure 2b is the overtaking USV’s encounter situation in which one USV overtakes the other USV from behind. The overtaking USV turns to the port or starboard side. The overtaken USV should keep the course. Figure 2c illustrates the cross-give-way USV encounter situation, in which the own USV has the responsibility to avoid the obstacle USV and should turn to the starboard side. Figure 2d reveals the cross-stand-on USV encounter situation. The own USV does not have the responsibility to avoid and should maintain the current course.

In order to ensure the safety of USV navigation, every USV has an inviolable ship domain. More specifically, the ship domain of each USV should maintain a mutually exclusive relationship [25]. As shown in Figure 3, *r* is the radius of the ship domain. Its size is affected by many factors such as the USV size, USV maneuverability, USV speed, and encounter situation [26]. In addition, *R* is the dynamic area, which indicates the maximum warning range of its own USV. When the obstacle USVs enter this range, the own USV should decide whether to change the course immediately to deal with the danger of collision.

### 2.2. USV Motion Variables

The three-degree-of-freedom (3DOF) movement of the USVs on the horizontal plane is shown in Figure 4. Y(N) points to True North, and X(E) points to True East. The red USV is the own USV ($USV_{U}$). The blue USV is the obstacle USV ($USV_{O}$). The yellow circle is the terminal for the USV navigation. $\left(x_{U},y_{U}\right)$ is the own USV position. $\left(x_{t},y_{t}\right)$ is the terminal position. $\varphi_{U}$ is the own USV course. $\varphi_{O}$ is the obstacle USV course. $\varphi_{t}$ is the absolute azimuth of the terminal and the own USV. ϕ is the relative azimuth of the target and the own USV. $V_{U}$ is the own USV speed. $V_{O}$ is the obstacle USV speed. *d* is the distance of the obstacle USV relative to the own USV. $\alpha_{O}$ is an absolute azimuth of the obstacle USV and the own USV. θ is a relative azimuth of the obstacle USV and the own USV. $V_{OU}$ is the relative speed of the obstacle USV relative to the own USV. $\varphi_{OU}$ is the relative speed direction of the obstacle USV relative to the own USV.

It is crucial to calculate the risk of collision during the encounter of USVs. It intuitively reflects the degree of hazard in the current situation. It also plays a key role in guiding the agent behaviors to avoid collisions reasonably. Therefore, as shown in Figure 5, two momentous parameters, DCPA and TCPA, are introduced to evaluate the collision risk. DCPA is the distance at the closest point of approaching, which reflects the urgency of the space when the own USV collides with the obstacle USVs. TCPA is the time to the closest point of approaching and reflects the urgency of the time with the obstacle USVs. In Figure 5, the $V_{OU}$ is the relative speed of the two USVs. γ is the difference between the relative position angle and the relative speed angle. Combining the parameters in Figure 4, the DCPA and TCPA calculation equations are as follows,
(1)$$\begin{array}{c} DCPA=d\sin(\gamma ) \end{array}$$
(2)$$\begin{array}{c} TCPA=d\cos\left(\gamma \right)/V_{OU} \end{array}$$

We can calculate the membership functions of DCPA and TCPA as follows,
(3)$$\begin{array}{c} u_{DCPA}=\left\{\begin{array}{cc} 1,\;\;\;\;\;\;\;\;\;\;|DCPA|\leq r & \\ 0.5-0.5\sin[\frac{\pi }{R-r}\times \frac{DCPA(R+r)}{2}],\;r<|DCPA|\leq R\\ 0,\;\;\;\;\;\;\;\;\;\;|DCPA|>R \end{array}\right. \end{array}$$

If TCPA > 0,
(4)$$\begin{array}{c} u_{TCPA}=\left\{\begin{array}{cc} 1,\;\;\;\;TCPA\leq t_{1} & \\ {\left[\frac{t_{2}-TCPA}{t_{2}-t_{1}}\right]}^{2},\;t_{1}<TCPA\leq t_{2}\\ 0,\;\;\;\;TCPA>t_{2} \end{array}\right. \end{array}$$

If TCPA ≤ 0,
(5)$$\begin{array}{c} u_{TCPA}=\left\{\begin{array}{cc} 1,\;\;\;\;|TCPA|\leq t_{1} & \\ {\left[\frac{t_{2}+TCPA}{t_{2}-t_{1}}\right]}^{2},\;t_{1}<\left|TCPA\right|\leq t_{2}\\ 0,\;\;\;\;,|TCPA|>t_{2} \end{array}\right. \end{array}$$
where,
(6)$$\begin{array}{c} t_{1}=\left\{\begin{array}{c} \frac{\sqrt{d_{1}^{2}+DCPA^{2}}}{V_{OU}},\;d_{1}\geq \left|DCPA\right|\\ 0,\;\;\;\;d_{1}<\left|DCPA\right| \end{array}\right. \end{array}$$
(7)$$\begin{array}{c} t_{2}=\left\{\begin{array}{c} \frac{\sqrt{d_{2}^{2}-DCPA^{2}}}{V_{OU}},\;d_{2}\geq \left|DCPA\right|\\ 0,\;\;\;\;d_{2}<\left|DCPA\right| \end{array}\right. \end{array}$$

Therefore, we can calculate the USV collision risk $u_{CRI}$ as,
(8)$$\begin{array}{c} u_{CRI}=\left\{\begin{array}{c} 0,\;\;\;\;\;\;\;u_{DCPA}=0\\ 0,\;\;\;\;\;\;\;u_{DCPA}\neq 0,u_{TCPA}=0\\ \max\left(u_{DCPA},u_{TCPA}\right),\;u_{DCPA}\neq 0,u_{TCPA}\neq 0 \end{array}\right. \end{array}$$

### 2.3. USV Mathematical Model

That the formulaic USV mathematical model can describe the movement characteristics of the USV in the virtual environment through a mathematical language provides the basis of the algorithm construction for the USVs’ autonomous collision avoidance algorithm. In this paper, the Norrbin USV mathematical model was applied to construct the autonomous collision avoidance algorithm for the USVs [27]. The equation of the Norrbin USV mathematical model is as follows,
(9)$$\begin{array}{c} \left\{\begin{array}{c} T\dot{\eta }+\eta +\alpha \eta^{3}=k\delta \\ \eta =\dot{\varphi } \end{array}\right. \end{array}$$
where *K*, α, and *T* are USV parameters, which reflect the USV following performance and gyration performance; these parameters can be identified for different USVs. δ is the angle of the rudder, which is the target value of the rudder angle control command. A bigger angle of the rudder means the course of the USV changes more drastically. φ is the course of the USV, and η is the rate of change of the course of the USV. It is worth noting that the USV mathematical model introduced into the collision avoidance algorithm is more realistic. Comparing the algorithm of the control of the course of the USV, it is equivalent to adding an additional constraint because the action of the course control by the rudder angle command will make the training process of reinforcement learning complex and realistic.

## 3. Reinforcement Learning Collision Avoidance Algorithm

Reinforcement learning is the theory of learning in interaction. This theory does not require any prior knowledge of humans, but it can complete numerous complex control tasks well [28]. Model-free reinforcement learning methods are very distinctive, and they do not explore the specific internal connections and hidden structures of objects [29], but are guided by interactive learning and aim at maximizing rewards. This feature makes reinforcement learning methods enjoy an excellent performance in complex and unknown environments because traditional algorithms have difficulty correctly and accurately representing the connections between agents and the environment. The autonomous collision avoidance process of the USVs is very suitable for the control mode of reinforcement learning. Firstly, the collision avoidance control process of USVs has high real-time requirements. For the time-varying environment, the reinforcement learning agent can make decisions immediately. Secondly, the essence of reinforcement learning is to solve the Markov decision process (MDP) question [30]. As shown in Figure 6, it contains three major elements: action, reward, and state. The agent perceives the current environment information and makes corresponding actions to change the environment. After the marine environment changes, an agent senses the marine environment again and obtains another reward. The USV collision avoidance control algorithm can apply this process conveniently without many additional complicated designs.

### 3.1. Reinforcement Learning and the Finite Markov Decision Process

As shown in Figure 7, the behavior of USV collision avoidance is transformed into a finite Markov decision process and a trajectory is obtained,
(10)$$\begin{array}{c} S_{0},A_{0},R_{1},S_{0},A_{1},R_{2},S_{2},A_{2},R_{3}\ldots \end{array}$$

The agent interacts with the environment at every step *t*. Since it is a finite Markov process, both $R_{t}$ and $S_{t}$ have a very clear probability distribution, and this can be expressed as move probabilities $p(s^{\prime },r|s,a)$. This can be described as the probability of $s^{\prime }$ and *r* appearing at step t+1 by choosing specific action *a* at current state *s*. Consequently, the subsequent $R_{t}$ and $S_{t}$ only depend on the $S_{t-1}$ and $A_{t-1}$ at the predecessor. Furthermore, the purpose of reinforcement learning is to maximize future rewards, and we can use $G_{t}=R_{t+1}+\gamma R_{t+2}+\gamma^{2}R_{t+3}+\ldots =\sum_{k=0}^{\infty }\gamma^{k}R_{t+k+1}$ to represent the future reward. γ is the reward decay, which is used to weaken the impact of future rewards on the current one.

In the theory of reinforcement learning, the value function is used to evaluate how excellent a state is. It is defined by future rewards and related to actions and states closely. The mapping from the state to the selection probability of each action is policy π. Therefore, the equation of the state-value function is as follows,
(11)$$\begin{array}{ll} v_{\pi }\left(s\right) & \overset{.}{\;=}E{}_{\pi }\left[G_{t}|S_{t}=s\right]\\ & =E{}_{\pi }\left[\sum_{k=0}^{\infty }\gamma^{k}R_{t+k+1}|S_{t}=s\right]\\ & =\sum_{a}\pi (a|s)\sum_{s^{\prime },r}p\left(s^{\prime },r|s,a\right)\left[r+\gamma v_{\pi }\left(s^{\prime }\right)\right] \end{array}$$

Similar, the action-value function can be expressed as,
(12)$$\begin{array}{c} \begin{array}{cc} \; & q_{\pi }\left(s,a\right)\overset{.}{\;=}E{}_{\pi }\left[G_{t}|S_{t}=s,A_{t}=a\right]\\ \; & \;\;\;=E{}_{\pi }\left[\sum_{k=0}^{\infty }\gamma^{k}R_{t+k+1}|S_{t}=s,A_{t}=a\right]\\ \; & \;\;\;=\sum_{s^{\prime }}p\left(s^{\prime },r|s,a\right)\left[r+\gamma \sum_{a^{\prime }}q_{\pi }\left(s^{\prime },a^{\prime }\right)\right] \end{array} \end{array}$$

Equation (Equation 11) is the Bellman equation, which expresses the connection between the current state value and subsequent states’ values. The reinforcement learning algorithm approximates the optimal policy $\pi_{*}$ based on the following optimal Bellman equation,
(13)$$\begin{array}{c} \begin{array}{cc} \; & q_{*}\left(s,a\right)=E\left[R_{t+1}+\gamma v_{*}\left(S_{t+1}\right)|S_{t}=s,A_{t}=a\right]\\ \; & \;\;\;=\sum_{s^{\prime },r}p\left(s^{\prime },r|s,a\right)\left[r+\gamma \max_{a^{\prime }}q_{*}\left(s^{\prime },a^{\prime }\right)\right] \end{array} \end{array}$$

### 3.2. Q Learning

Model-free reinforcement learning method can be divided into the policy gradient method [31,32] and value-based method. In the value-based method, the temporal-difference (TD) method is the most novel [33]. The off-policy algorithm, Q learning, in the TD method has a very outstanding performance [34]. The definition of the Q learning algorithm is as follows,
(14)$$\begin{array}{c} Q\left(S_{t},A_{t}\right)\leftarrow Q\left(S_{t},A_{t}\right)+\alpha \left[R_{t+1}+\gamma \max_{a}Q\left(S_{t+1},a\right)-Q\left(S_{t},A_{t}\right)\right] \end{array}$$
where α is the learning rate, which is used to control the speed of learning. In the off-policy algorithm, there are two policies: behavior policy and target policy. The decision sequences used for agent learning have nothing to do with what action the agent takes. The off-policy algorithm will introduce an inevitable error, but the learning speed is faster.

### 3.3. Deep Q Network

The DQN algorithm approximates the value function by introducing neural network parameter θ. In this way, it no longer needs to maintain a huge Q value table. Such a neural network is called a deep Q network, and the DQN algorithm update process is as shown in Figure 8. Before the DQN algorithm was proposed, people were generally not very optimistic about the approximating of neural networks in reinforcement learning. The reinforcement learning algorithm is unstable when a neural network is used to approximate, which due to the sampling not being independent, the rewards being delayed and sparse, and the noise being general. Furthermore, the data distribution of reinforcement learning is changing all the time, which is very unfavorable for the learning of neural networks. There had been many studies on these issues before DQN was proposed [35,36,37]. However, none of them had come up with an excellent solution to these problems. The historic breakthrough was the experience replay buffer and the two-neural-network architecture of the DQN algorithm, which disrupts the temporal correlation between samples well. In every step, the deep reinforcement learning agent interacts with the environment, and a transition $\left(s,a,r,s^{\prime }\right)$ will be generated through the behavior policy and stored in the experience replay buffer for learning. Because the basic algorithm of DQN is bootstrapping update mode, at each step, the agent samples a certain number of transitions from the experience replay buffer and learns. The DQN algorithm has two neural networks with the same architecture, but different parameters to output the Q-evaluate and Q-target. Equation (Equation 15) is the loss function, and the network parameters of the evaluate network are updated every step through the gradient of this loss function (Equation (Equation 16)), which is used to output the Q-evaluate. The target network copies the parameters of the estimate network every *N* step and outputs the Q-target.
(15)$$\begin{array}{c} L\left(\theta \right)=E\left[{\left(r+\gamma \max_{a}Q\left(s^{\prime },a^{\prime };\theta^{-}\right)-Q\left(s,a;\theta \right)\right)}^{2}\right] \end{array}$$
(16)$$\begin{array}{c} \frac{\partial L(\theta )}{\partial \theta }=E\left[\left(r+\gamma \max_{a}Q\left(s^{\prime },a^{\prime };\theta^{-}\right)-Q\left(s,a;\theta \right)\right)\frac{\partial Q(s,a;\theta )}{\partial \theta }\right] \end{array}$$

### 3.4. Collision Avoidance Algorithm with RL

For the autonomous real-time collision avoidance problem of USVs, a Markov chain can be formed directly. Then, the collision avoidance problem can be transformed into a Markov decision problem, and it can use the reinforcement learning method to solve the problem. In this paper, the reinforcement learning agent can achieve the task of navigating to the terminal and avoiding obstacles.

First, the state of the reinforcement learning USV agent was designed. The state space should fully reflect the current navigation condition and the surrounding marine environment of the USVs. There are many sensors when the USV is navigating at sea, and much accurate USVs information can be obtained through these sensors. The design of the state space in this paper is as follows,
(17)$$\begin{array}{c} S=[\varphi_{U},{\dot{\varphi }}_{U},\delta_{U},V_{U},\varphi_{t},d_{1},\theta_{1},\varphi_{O1},V_{O1},d_{2},\theta_{2},\varphi_{O2},V_{O2}\ldots ] \end{array}$$

Such a state space design can fully reflect the various information of USV movement required for autonomous collision avoidance. Including the own USV navigation information, terminal information, and obstacle USVs’ information, the state space is very important for the implementation of algorithm. The design of $\varphi_{U}$ and $\varphi_{t}$ reflects the deviation of the own USV’s course from the course of navigating towards the terminal. The agent can learn that the action towards the terminal is better through continuous learning. ${\dot{\varphi }}_{U}$ and $\delta_{U}$ can make not only the description of the current navigation state of the own USV more accurate, but also are an essential basis for calculating the next time course of the own USV through the USV mathematical model in the simulation. *d* is the distance between the own USV and the obstacle USVs. *d*, θ, and $\varphi_{O}$ describe the information of the obstacle USVs and provide a basis for the agent to learn collision avoidance behaviors.

In second place is the design of the action in the autonomous collision avoidance algorithm. The action space needs to reflect the motion characteristics and maneuverability of the USVs fully. Since the DQN algorithm can only deal with the problem of discrete actions, the design of the action space in this paper is as follows,
(18)$$\begin{array}{c} A=[\Delta \delta_{U1},\Delta \delta_{U2},\Delta \delta_{U3},\Delta \delta_{U4}\ldots \Delta \delta_{Un}] \end{array}$$
(19)$$\begin{array}{c} \delta_{t\arg et}=\Delta \delta_{Uk}+\delta_{current} \end{array}$$

The design of the action space should conform to the actual situation of the USVs so that the training of the agent in the simulation is more realistic and usable. For the action space, each action $\Delta \delta_{Uk}$ represents the change in rudder angle relative to the last state of the own USV. Indeed, the size and quantity of $\Delta \delta_{Uk}$ should be based on the actual situation. $\delta_{t\arg et}$ is the target rudder angle, which is the expected rudder angle in the next step. Under the premise that the target rudder angle will not exceed the maximum rudder angle of the USVs, the USVs with better maneuverability and the action of the rudder angle change should be set larger. This way of the action space design makes the trained agent more realistic.

Finally, the design of the reward functions is the most intricate and challenging part of the algorithm design. Guiding the agent to achieve the expected terminal quickly and efficiently requires the subtle planning of the reward function. Exquisite reward function setting can not only make the convergence faster, but also make the trained agent better. A terrible reward function setting may cause the agent to fall into a local optimum or even fail to achieve an approximate convergence effect. Therefore, the general reward functions are divided into the final reward of the task and the stage reward of guiding in the middle. An ideal reward function design should only have end rewards; however, this makes the reinforcement learning questions become the problem of sparse rewards, which makes the training more difficult. Especially for the agent with a USV mathematical model, the design only with the final rewards makes it difficult to complete the collision avoidance task and reach the terminal because the agent may obtain too little information about the final reward in the training. However, adding too many stage rewards will interfere with the impact of the final reward on the environment. If the design is unreasonable, it will easily cause the algorithm to fail to approximate the convergence. A reasonable ratio between the final rewards and the stage rewards can make the training of the agent faster and better. Therefore, the reward functions were set as follows.

The first is the design of the final rewards:(1)The terminal is the position where the USV is expected to arrive at the end, and the terminal reward can guide the agent to the terminal position well. Therefore, the reward for reaching the terminal position is,
(20)$$\begin{array}{c} R_{goal}=k_{goal}r_{goal},\;\sqrt{{\left(x_{U}-x_{t}\right)}^{2}+{\left(y_{U}-y_{t}\right)}^{2}}<2r; \end{array}$$(2)The most considerable thing in the collision avoidance of USVs is avoiding collision with the obstacle USVs. Therefore, the design of the collision reward can constrain the collision avoidance behavior strongly. The reward design for collisions is,
(21)$$\begin{array}{c} R_{\mathrm{collision}}=k_{collision}r_{collision},\;\sqrt{{\left(x_{U}-x_{O}\right)}^{2}+{\left(y_{U}-y_{O}\right)}^{2}}<2r; \end{array}$$(3)It is expected that the algorithm for the autonomous collision avoidance of USVs conforms to the actual COLREGs. As a consequence, it needs to make positive rewards for actions that follow the COLREGs and earn negative rewards for actions that are against the COLREGs. This kind of reward function can better constrain the USV behaviors following the COLREGs in different encounter situations. When the obstacle USV enters the dynamic area and with the danger of collision, according to the COLREGs, the actions of unreasonable rudder angle changes should be punished. Therefore, the reward function of the COLREGs is as follows.When $\sqrt{{\left(x_{U}-x_{O}\right)}^{2}+{\left(y_{U}-y_{O}\right)}^{2}}<R$,
(22)$$\begin{array}{c} \left\{\begin{array}{c} R_{COLREGs}=0,\;\;\;\;\;following\;the\;COLREGs\\ R_{COLREGs}=k_{COLREGs}u_{CRI},\;else \end{array}\right. \end{array}$$When $\sqrt{{\left(x_{U}-x_{O}\right)}^{2}+{\left(y_{U}-y_{O}\right)}^{2}}\geq R$,
(23)$$\begin{array}{c} R_{COLREGs}=0 \end{array}$$$k_{COLREGs}$ is the scale factor, and $u_{CRI}$ is the collision risk.

The stage reward is as follows:(1)At each step, the agent will reach another state and obtain some specific stage rewards. This makes the agent able to obtain plentiful rewards during the exploration process, which guide the agent to achieve the terminal better. Therefore, certain rewards for the course $\varphi_{U}$ were designed. The reward of the course is,
(24)$$\begin{array}{c} R_{\varphi }=k_{\varphi }\left(\varphi_{k}-\left|\varphi_{U}-\varphi_{t}\right|\right) \end{array}$$
where $k_{\varphi }$ is a scale factor, and $\varphi_{k}$ is a value distinguishing positive and negative rewards in the training environment;(2)It is desirable that the course of the USV be closer to $\varphi_{t}$, so the rewards of the course changes are as follows.If ϕ becomes smaller,
(25)$$\begin{array}{c} R_{\Delta \varphi }=r_{smaller} \end{array}$$If ϕ becomes bigger,
(26)$$\begin{array}{c} R_{\Delta \varphi }=r_{bigger} \end{array}$$If ϕ remains unchanged,
(27)$$\begin{array}{c} R_{\Delta \varphi }=\left\{\begin{array}{c} r_{equal1},\;\phi =0\\ r_{equal2},\;\phi \neq 0 \end{array}\right. \end{array}$$The total reward function can be written as,
(28)$$\begin{array}{c} R_{all}=R_{goal}+R_{collision}+R_{COLREGs}+R_{\varphi }+R_{\Delta \varphi .} \end{array}$$

The complete training process is shown in Figure 9. The left part of this figure is the real-time simulation of the marine environment. At a step *t* of one episode, various real-time information of the own USV and the obstacle USVs is input into the evaluate network and target network. The target network outputs the value of all the actions that the USV can make in the current situation. Subsequently, the USV uses the ε greedy strategy to select the action based on the output result of the neural network. The choice of keeping the rudder angle or changing the rudder angle to avoid the obstacle USVs should be decided. Then, the marine environment changes and the next loop from perception to control are carried out.

## 4. Improvement of the DQN Algorithm

### 4.1. D3QN Algorithm

Such overestimation of the action value is general, which is attributed to the unknown actual action value during the process of agent training by the DQN algorithm. This may affect the performance of the algorithm or even make the agent fall into a local optimum [38]. The optimistic estimate is not terrible for learning [39]; however, inconsistently optimistic estimation will lead to bad performance because overestimation does not always happen on the actions that the agent should choose and learn more. The overestimation will lead to errors in the learning process and even failure to converge to the optimal policy. Because of the update method of bootstrapping, the state would have already been estimated, would significantly affect the entire training environment, and would lead to the unsatisfactory performance of the algorithm.

As shown in Equation (Equation 15), in the DQN algorithm, the action of the own USV is selected through the target network, and the estimate of the value of this action is also output through this network. Such an update model inevitably becomes the root of overestimation [40]. Therefore, decoupling the outputs of the action choice and the outputs of the action value estimate in the two neural networks is a terrific solution. The selection of the action is not output through the target network, but is based on the output of the estimated network. Therefore, the impact of the overestimated action value is reduced. The update equation of DQN is as Equation (Equation 29), and the update equation of the double-DQN is as Equation (Equation 30).
(29)$$\begin{array}{c} Y_{t}^{DQN}=r+\gamma \max_{a}Q\left(s^{\prime },a;\theta^{-}\right) \end{array}$$
(30)$$\begin{array}{c} Y_{t}^{Double\_DQN}=r+\gamma Q\left(s^{\prime },\underset{a}{\arg\max}Q\left(s^{\prime },a;\theta \right);\theta^{-}\right) \end{array}$$

There are many methods of the innovation of the neural network architecture in reinforcement learning, and the dueling architecture is an extremely effective and concise improvement [41]. As shown in Equation (Equation 31), it factors the action-value function as a state-value function and an advantage function. If an action-value function is very high, this is because the agent is in a great state or the agent chooses an excellent action in a normal state. Therefore, dueling-DQN will achieve better results when there are many similar action-value functions. The dueling network has two streams, which are used to output the state value function and the advantage function. The state-value function describes the level of a state, and the advantage function describes the value of each action selected based on this state. This improved method changes the architecture of the algorithm very little, but it makes the algorithm more robust. The dueling network architecture is as shown in Figure 10.
(31)$$\begin{array}{c} Q(s,a;\theta ,\alpha ,\beta )=V(s;\theta ,\beta )+A(s,a;\theta ,\alpha ) \end{array}$$

### 4.2. Category-Based Exploration Method

Exploration and exploitation comprise a pair of contradictory issues that are for all reinforcement learning methods. This is a considerable problem when dealing with uncertain systems that need to take into account the two points of identification and control. In this paper, it was hoped that the agent can not only explore a broader range of USV situations in small training times, but also make full use of the existing valuable transitions to converge to the optimal strategy. Therefore, the ε greedy strategy was adopted to balance exploration and exploitation. In each step, with the possibility of ε, the action that the agent takes is based on the maximum action value of the neural network output. Besides, it also chooses the random action with a probability of 1−ε. In the initial stage, the parameters of the neural network are far from the ideal result. Therefore, a smaller ε makes the agent perform more random exploration behaviors in the initial stage. The length of the ε increasing process is related to many factors such as the difficulty for USVs to reach the terminal and the depth and width of the neural networks.

Exploration has always been an essential problem in reinforcement learning [42]. This study aimed at the exploration problem of the agent, considering the characteristics of the USV agent in the training, improving the traditional reinforcement learning count-based exploration method, a technique based on category counting to optimize the exploration [43,44]. The states the agent had not explored or had explored less may have more attraction. Therefore, in the tabular reinforcement learning problem, the times each state appears are counted [45]. If a state appears once, add one to the number of visits to it. For the USV training environment, it is very unrealistic to record every state. Because the number of states is very huge, the number of occurrences of each state is very tiny, and therefore, this counting method is very inappropriate in several hundred thousand training steps. For the USV training environment, recording similar states as one category is very suitable. Counting by category would be more feasible, and the way of classification based on some states of the USV is as follows,
(32)$$\begin{array}{c} S=[\varphi_{U},\varphi_{t},d_{1},\theta_{1},\varphi_{O1},d_{2},\theta_{2},\varphi_{O2}\ldots ] \end{array}$$

Similar angles are divided into one category. The distance between obstacle USVs and the own USV is divided into far, medium, and close. In this way, the number of states is significantly reduced, and we can use the improved method based on the approximate count-based exploration method feasibly. In this paper, an additional reward $R_{\exp lore}$ to incentivize the exploration of unknown areas was designed. As shown in Equation (Equation 33), σ is the bonus coefficient, which is used to adjust the impact of the additional reward on training. $d_{t}$ is used to prevent the denominator from being zero. n(S) is the number of occurrences of a category. The additional reward of the categories that appear more frequently will be smaller. Therefore, as the training progresses, the attractiveness of the categories that are explored more becomes less. This improved method will improve the algorithm effect.
(33)$$\begin{array}{c} R_{\exp lore}=\frac{\sigma }{\sqrt{n(S)}+d_{t}} \end{array}$$
(34)$$\begin{array}{c} R_{improve}=R_{all}+R_{\exp lore} \end{array}$$

Combining all the optimization details above, the RLCA algorithm update code is shown in Algorithm 1.
**Algorithm 1** RLCA algorithm codeInitialize the training environmentInitialize the replay memory buffer to capacity *D*Initialize the evaluate network with random weight θInitialize the target network with random weight $\theta^{-}=\theta$For episode = 1, M doInitialize the initial position of the own USV and the obstacle USVsWhile trueupdate the training environmentWith probability ε, select USV action $a_{t}\in A_{t}$Otherwise, select USV action $a_{t}=\underset{a}{\arg\max}Q\left(s^{\prime },\underset{a}{\arg\max}Q\left(s^{\prime },a;\theta \right);\theta^{-}\right)$Execute action $a_{t}$ in the training environment, and obtain $s_{t+1}$Obtain reward $R_{all}=R_{goal}+R_{collision}+R_{COLREGs}+R_{\varphi }+R_{\Delta \varphi }$ via maneuvering and the COLREGsObtain the $s_{t+1}$ category, and add one in n(S) by the method of category-based explorationObtain a reward $R_{\exp lore}=\frac{\sigma }{\sqrt{n\left(S\right)+d_{t}}}$ based on category-based explorationObtain the total reward $R_{improve}=R_{all}+R_{\exp lore}$Store transition $\left(s_{t},a_{t},r_{t},s_{t+1}\right)$ in replay memory buffer *D*Sample the random minibatch of transitions $\left(s_{j},a_{j},r_{j},s_{j+1}\right)$ from *D*Obtain $y_{i}=\left\{\begin{array}{c} r_{j},\;\;\;\;\;\;\;if\;j+1\;is\;the\;terminal\\ r_{j}+\gamma \max_{a^{\prime }}Q\left(s_{j},a^{\prime },\theta^{-}\right),\;otherwise \end{array}\right.$Update the evaluate network parameters θ with gradient descentIf the number of steps reaches the update step *N*update the target network with weight $\theta^{-}=\theta$End ifThe number of steps plus 1End whileEnd whileReturn the weight $\theta^{*}=\theta^{-}$

### 4.3. Some Algorithm Details

Since this paper aimed at USVs with outstanding maneuverability, there were much big rudder angle changes under the design of the action space based on the USV mathematical model. Within a few thousand steps at the beginning of the training, having large rudder angle change actions, and a lack of understanding of the environment, there is a large number of USV turning behaviors. To ensure the reliability and authenticity of the algorithm, it is not desirable to delete the USV mathematical model or reduce the maximum value of the change in the rudder angle. Therefore, it will accumulate a certain number of transitions in the experience replay buffer about the agent close to the starting point with turning around behaviors. However, at the end stage of the agent training, these transitions will affect the training. Undeniably, it is desirable that the agent achieve convergence to the optimal strategy at the end stage. So, it needs as much as possible to increase the probability of sampling the transitions which have more values. Therefore, an appropriate size of the experience replay buffer was set. When the experience replay buffer is full, new transitions will replace part of the earlier transitions. Therefore, in the end-stage, the agent can sample more transitions that it wants to learn, as well as can speed up the convergence of the optimal policy. This can improve the performance of collision avoidance, increase the stability of the algorithm, and obtain a higher average reward.

## 5. Experiments

Based on the analysis of the collision avoidance process and the design of the RLCA algorithms, in this section, the training process of the reinforcement learning agent for the USVs is designed. A better average reward and collision avoidance result for the RLCA algorithm in the USV simulation environment is shown. This experiment was conducted under the Windows10 system, Intel Xeon Gold 5218 CPU, 64 GB memory, 2 × NVIDIA GeForce RTX 2080 Ti GPU, and Python 3.6. The collision avoidance simulation environment was under Canvas in Tkinter. The framework of the algorithm was Tensorflow 1.15.

### 5.1. Training and Parameter Setting

Figure 11 shows the sea area near Dalian Maritime University, and the area of the red square in this figure is the sea area for USV collision avoidance training. As shown in Figure 12, it is an enlarged view of the training marine area, for which the size was set to 1000 m × 1000 m. When the own USV leaves this range, it is regarded as the termination of this episode. The red USV is the own USV; the blue USVs are the obstacle USVs; the yellow ball is the terminal. The initial and terminal positions of the own USV remain unchanged in each episode. The initial position of the obstacle USV is set in every $\mu =1^{\circ }$ direction on the dotted circle of the picture. A total of 360 obstacle USV initial positions were set, starting from a random initial position in each episode. Furthermore, in each episode, the own USV can change its rudder angle, and the obstacle USVs do not change their direction or speed.

As shown in Figure 13, “Lan Xin” USV was the object in this experiment, and its parameters are shown in Table 1. Therefore, in this simulation environment, the radius of the ship domain was 15 m, and the speed of the own USV remained constant at 5.6 m/s. The hyperparameter setting of reinforcement learning is shown in Table 2. A total of 2000 episodes were trained, and the batch size was 600. The size of the experience replay buffer was 350,000. During the training process in this experiment, when the upper limit of the experience replay buffer was reached, 50% of the early transitions in the experience replay buffer were replaced. In order not to have the USV turning in circles all the time in one episode, the upper limit of the number of steps in each episode to 500 steps was set. If the limit were exceeded, it would enter the beginning of the next episode automatically. The target network copies the parameters of the evaluate network every 5360 steps to realize the actual learning process of the agent. The learning rate was set to 0.0001 because a lower learning rate would make the training process more stable. According to the formula $\gamma \approx 0.1^{1/t}$, it is necessary that the reward after *t* steps still accounts for about 10% of the Q value in the current step. Therefore, the discount factor was set to 0.99, and it made the foresight of the agent in this experiment reach about 230 steps. A larger γ means that the rate of convergence will be slower and the agent will be more susceptible to noise. However, in general, a low discount factor will make the convergence effect of the algorithm more stable. This experiment adopted a changing greedy value. At the beginning, the greedy value was set to 0; therefore, the agent would choose the action at random entirely. Then, it increased to 0.9 after 30,000 steps. In this way, the agent can select more exploratory behaviors in the early stage to find more unknown states for better learning later. Exploratory behaviors make the agent have a fuller understanding of the entire environment, which is very beneficial to the convergence process in training. In the later stage, the greedy value was very high. With a very high probability, the output of the neural network was used to perform the optimal chosen under the current training result so that the current optimal policy can be improved continuously.

The state space of the single-obstacle-USV collision avoidance experiment is as follows,
(35)$$\begin{array}{c} S=[\varphi_{U},{\varphi_{U}}^{\prime },\delta ,V_{U},\varphi_{t},d_{1},\theta_{1},\varphi_{O1},V_{O1}] \end{array}$$

Since the DQN algorithm can only solve the problem with discrete actions and considering that the USV has very outstanding maneuverability, more actions in the action space were designed. Therefore, in the training process, it was more difficult to explore the terminal position. This made the rising stage of the average reward appear later. The action space of the single-obstacle-USV collision avoidance experiment is as follows,
(36)$$\begin{array}{c} A=[-5^{\circ },-4^{\circ },-3^{\circ },-2^{\circ },-1^{\circ },0^{\circ },+1^{\circ },+2^{\circ },+3^{\circ },+4^{\circ },+5^{\circ }] \end{array}$$

The rewards functions were set as follows,
(37)$$\begin{array}{c} R_{goal}=+10,000 \end{array}$$
(38)$$\begin{array}{c} R_{\varphi }=200\left(6-\left|\varphi_{U}-\varphi_{t}\right|\right) \end{array}$$
(39)$$\begin{array}{c} R_{COLREGs}=-10u_{CRI} \end{array}$$
(40)$$\begin{array}{c} R_{\mathrm{collision}}=-2000 \end{array}$$

To prevent the agent from obtaining a higher reward value by constantly changing its course, the negative reward of $R_{\Delta \varphi }$ was designed to be larger. Therefore, the design of $R_{\Delta \varphi }$ was as follows.

If ϕ becomes smaller,
(41)$$\begin{array}{c} R_{\Delta \varphi }=10 \end{array}$$

If ϕ becomes bigger,
(42)$$\begin{array}{c} R_{\Delta \varphi }=-15 \end{array}$$

If ϕ remains unchanged,
(43)$$\begin{array}{c} R_{\Delta \varphi }=\left\{\begin{array}{c} 10,\;\phi =0\\ -15,\;\phi \neq 0 \end{array}\right. \end{array}$$

The additional reward $R_{\exp lore}$ is shown as Equation (Equation 44). When comparing with the other algorithm, without this kind of reward, we needed to remove the exploratory reward to compare in the same context.
(44)$$\begin{array}{c} R_{\exp lore}=\frac{0.005}{\sqrt{n(S)}+0.0001} \end{array}$$

For the selection of the neural network architecture in this experiment, there were 2 hidden layers with 512 neurons in each layer. The number of neurons in the input layer and output layer was equal to the number of states and actions in their space. The activate function adopted was the ReLU function, and the optimization algorithm used RMSProp, which has good efficiency and stability.

The training effect of different training stages during the training process is shown in Figure 14.

Figure 14a shows the effect of training in about 20 episodes. There were many turning behaviors, and the performance of collision avoidance was far from the ideal effect.

Figure 14b,c is the effects of training in about 50 and 80 episodes. The agent began to explore more unknown states. Figure 14d,e is the effects of training in about 100 and 200 episodes. The exploration process of the agent became more direct, and there were no longer turning behaviors at the beginning of each episode. The extra exploratory reward reflected the key role of stage rewards. It guided the agent to explore the terminal effectively. Compared with the aimless exploration under sparse rewards, exploring about 200 episodes to the terminal was pretty fast. As shown in Figure 15, the effect of deleting the stage rewards was the USV falling into turning behaviors.

As shown in Figure 14f, in about 400 episodes, the effect of the USV navigation was closer to the end. Figure 14g,h is the effects of training in about 500 and 520 episodes. The transitions of reaching the terminal already appeared. Through learning, the behavior of the USV would be closer to a straight line that points to the terminal. However, these behaviors would lead to more collisions.

Figure 14i,j is the effects of training in about 700 and 1000 episodes. With more transitions of the collisions and COLREGs, the agent learned how to avoid collisions reasonably and effectively. As shown in Figure 14k,l, when learning about 1800 and 2000 episodes, the collision avoidance effect of the USV was relatively ideal, and the goal of USV autonomous collision avoidance was well achieved.

The comparison of the RLCA algorithm, D3QN algorithm, and DQN algorithm is shown in Figure 16. In this experiment, 10 groups of random number seeds were selected and the results were averaged in every episode. To make the contrast effect more pronounced, in this figure, the average reward of every 40 episodes was averaged and recorded as 1 value for drawing so that the contrast effect and the trend of changes could be made more explicit without destroying the original effect. The extra exploratory reward of the RLCA algorithm needed to be removed so that it could be compared with other algorithms in the same environment. In Figure 16b, it can be seen that the RLCA algorithm could obtain a higher average reward. At about 500 episodes, there was a stage where the average reward dropped. This was because in the early stage, the transitions of the collisions and COLREGs in the experience replay buffer were limited. Through continuous learning, the collision avoidance effect would be closer to the obstacle USVs. The agent did not learn how to deal with these situations. Therefore, the unfamiliar behaviors would cause the average reward to decrease. As more collision transitions are learned, the average reward value would continue to converge to a higher and stable stage. The average rewards with the RLCA algorithm rose slightly more slowly in the early stages, because the intelligent agents were driven by curiosity to perform more exploratory behaviors. However, in the later stage, it could be more stable and obtain higher rewards. Figure 16a is the average reward of the RLCA algorithm with the confidence region. By drawing the range of the maximum and minimum average reward value in each of the 10 groups of random number seeds, the approximate confidence region of the algorithm could be obtained, which well reflected the performance of the algorithm.

The trained reinforcement learning USV autonomous collision avoidance agent was applied to several encounter conditions. Figure 17 is the head-on encounter. Figure 17a is the collision avoidance effect of the RLCA algorithm in the head-on encounter. Figure 17b is the change in distance between the two USVs in Figure 17a. The own USV avoided the obstacle USV and navigated to the terminal successfully. The closest distance to the obstacle USV was 69.44 m. Figure 17c is the change in course of the own USV. The initial course of the own USV was 45${}^{\circ }$, and it kept this value when there was no avoidance behavior. Then, it changes to about 100${}^{\circ }$ and slowly recovered to the initial course when the encounter was over. Figure 17d is the θ of the own USV. The initial θ of the own USV was 0${}^{\circ }$. When the own USV avoided the obstacle USV, θ changed to about 180${}^{\circ }$. Because plus or minus 180 is a critical value, in Figure 17d, the value of θ kept going up and down.

Figure 18 is a cross encounter. Figure 18a is the collision avoidance effect of the RLCA algorithm in the cross encounter. Figure 18b is the change in distance between the two USVs in Figure 18a. The own USV avoided the obstacle USV and navigated to the terminal successfully. The closest distance to the obstacle USV was 69.34 m. Figure 18c is the change in course of the own USV. The initial course of the own USV was 45${}^{\circ }$, and the process was similar to the head-on situation. The course changed to about 110${}^{\circ }$ and slowly recovered to the initial course when the encounter was over. Figure 18d is the θ of the own USV. The initial θ of the own USV was 15${}^{\circ }$. When the own USV avoided the obstacle USV, θ changed to about −160${}^{\circ }$.

Figure 19 is another cross encounter. Figure 19a is the collision avoidance effect of the RLCA algorithm in the cross encounter. Figure 19b is the change in distance between the two USVs in Figure 19a. The own USV avoided the obstacle USV and navigated to the terminal successfully. The closest distance to the obstacle USV was 101.00 m. Figure 19c is the change in course of the own USV. The initial course of the own USV was 45${}^{\circ }$, and the process was similar to the head-on situation. The course changed to about 100${}^{\circ }$ and slowly recovered to the initial course when the encounter was over. Figure 19d is the θ of the own USV. The initial θ of the own USV was 25${}^{\circ }$. When the own USV avoided the obstacle USV, θ changed to about −140${}^{\circ }$.

Figure 20 is the overtaking encounter. Figure 20a is the collision avoidance effect of the improved algorithm in the overtaking encounter. Figure 20b is the change in distance between the two USVs in Figure 20a. The own USV avoided the obstacle USV and navigated to the terminal successfully. The closest distance to the obstacle USV was 46.37 m. Figure 20c is the change in the course of the own USV. The initial course of the own USV was 45${}^{\circ }$, and the process was similar to the head-on situation. Because of the obstacle USV interference with the own USV in the position near the terminal, the course did not recover to the initial course when the encounter was over. Figure 20d is the θ of the own USV. The initial θ of the own USV was 0${}^{\circ }$. When the own USV avoided the obstacle USV, θ changed to about 155${}^{\circ }$.

It can be seen that the trained agent could cope with various encounters well. The agent learned the behaviors of navigating to the terminal when the obstacle USV was far away from the own USV and the behaviors of avoiding the obstacle USV when the obstacle USV was closer to the own USV.

### 5.2. The Experiments on Multi-USV Collision Avoidance

As an equally significant part, the simulations of multiple USVs were also designed. A variety of marine encounters were designed. The USV agent avoided the obstacle USVs and reached the terminal successfully.

Figure 21 shows Simulation Environment 1. The red circle is the starting point of the own USV, and the yellow circle is the terminal. There were two obstacle USVs in this marine simulation environment, and they are represented by blue and violet circles. The own USV first encountered the blue obstacle USV in the crossing encounter situation, and the own USV turned right under the requirements of the COLREGs to avoid the obstacle USV successfully. Then, it was ready to recover to navigate to the terminal; however, at this time, the violet obstacle USV hindered the current behavior in a situation of overtaking. Therefore, the own USV performed a small right turn behavior until it was judged that the violet USV was not in danger of collision. Then, the own USV continued to the terminal.

Figure 22 shows another simulation environment. The settings of USV were similar to Environment 1. The own USV first encountered the blue obstacle USV in the head-on encounter situation, and the own USV turned right under the requirements of the COLREGs to avoid the obstacle USV successfully. Then, the own USV encountered the violet obstacle USV, but there was no danger of collision. Therefore, the own USV judged that the violet USV was not in danger of collision. Then, the own USV continued to the terminal.

## 6. Conclusions and Prospects

Based on the RLCA algorithm, in this paper, an autonomous collision avoidance method for USVs was designed. It does not require introducing many human experiences to better solve more complex collision avoidance problems of USVs. This training process fully considers the maneuverability and COLREGs in USV collision avoidance, and the Norrbin USV mathematical model was added to restrict the training effectively. Furthermore, the traditional method of storing transitions is not reasonable due to the many turning behaviors of the own USV’s in the early stage. Therefore a transition storage method suitable for USV training was designed. In addition, combining the count-based exploration method in traditional reinforcement learning and the characteristics of USVs, a category-based exploration method for USVs was proposed, and a good result was obtained. A comparison of the RLCA algorithm, D3QN algorithm, and DQN algorithm was made, and the RLCA algorithm had a higher average reward value. Finally, the agent trained by the RLCA algorithm was applied to face the typical encounter situations, and a good USV collision avoidance effect was achieved.

Taken together, this work illustrated the power of the reinforcement learning method in USV collision avoidance sufficiently; however, there are still many issues that need to be explored further. In future works, the following aspects will be carried out:(1)More factors in the training process should be considered, such as environmental interference and a more accurate USV mathematical model. This would make the trained agent more realistic;(2)The sampling method and network architecture will be further improved to obtain better training results;(3)Multi-agent reinforcement learning methods will be considered coping with the collaborative collision avoidance of multiple USVs;(4)Although reinforcement learning methods have been used in many research works on USV collision avoidance, their applications have been limited to experiments in simulation environments. In the subsequent works, the proposed autonomous collision avoidance algorithm for USVs will be applied to and experimented on real USVs. As shown in Figure 23, this will be the reinforcement learning autonomous collision avoidance control structure of the Lan Xin USV. It includes many modules for information collection, collision avoidance decision-making, and motion control, which maintain the operation of the collision avoidance system. It is certain that the applications of collision avoidance for USVs based on reinforcement learning algorithms will be bound to become more and more widespread;(5)The COLREGs are behavioral constraints for vessels commanded by humans. Because there are still few unmanned ships at sea today, the USVs trained under the COLREGs can coexist more easily with vessels commanded by humans. This is why the COLREGs were chosen in this paper. This is an attempt and not necessarily the best rule. How to design more suitable collision avoidance rules for USVs will be studied.

## Figures and Tables

**Figure 1 sensors-22-02099-f001:**
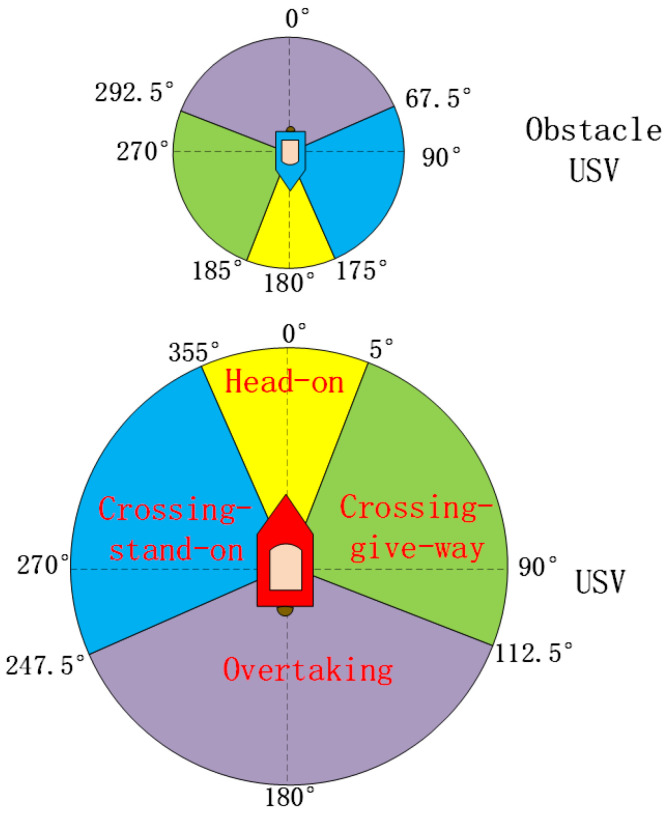
Encounter angle division of USVs under the COLREGs.

**Figure 2 sensors-22-02099-f002:**
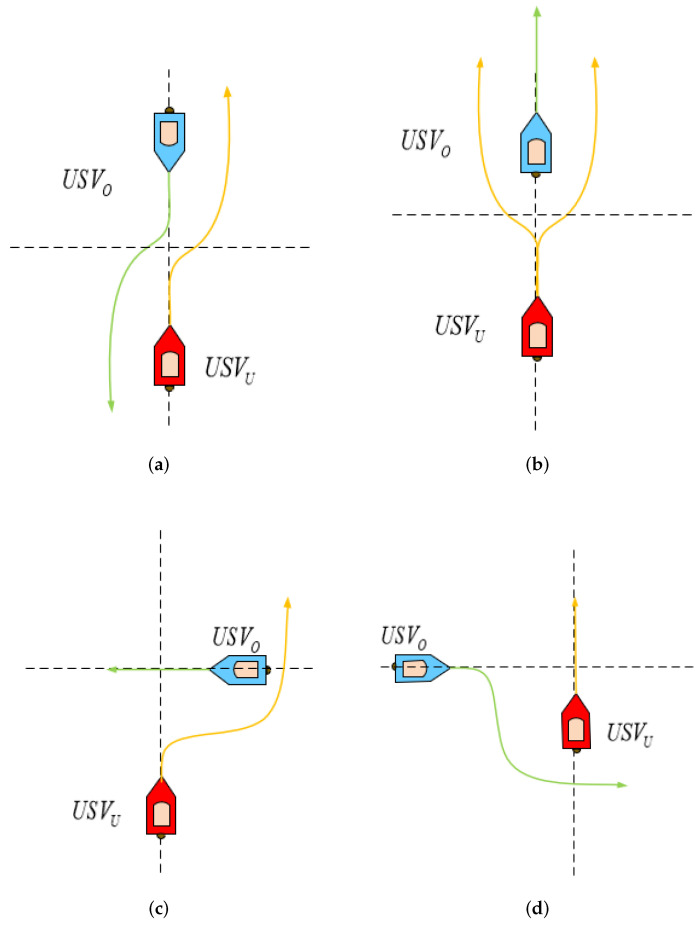
The encounter situations of USVs under the COLREGs. (**a**) Head-on. (**b**) Overtaking. (**c**) Crossing-give-way. (**d**) Crossing-stand-on.

**Figure 3 sensors-22-02099-f003:**
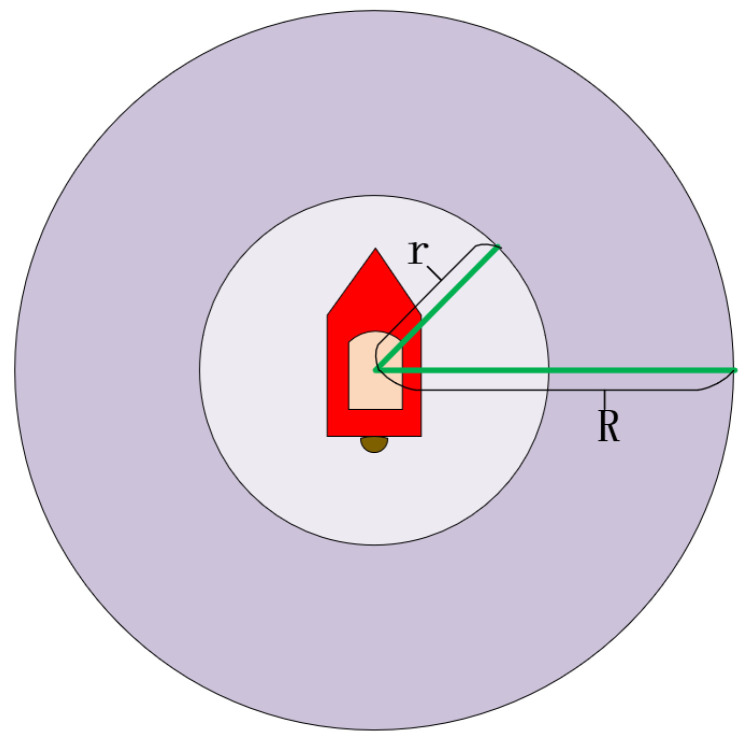
Ship domain and dynamic area.

**Figure 4 sensors-22-02099-f004:**
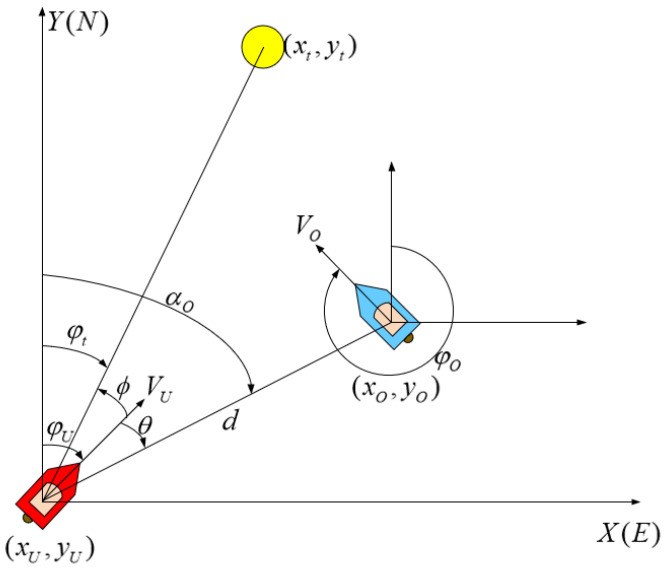
Schematic diagram of USV navigation.

**Figure 5 sensors-22-02099-f005:**
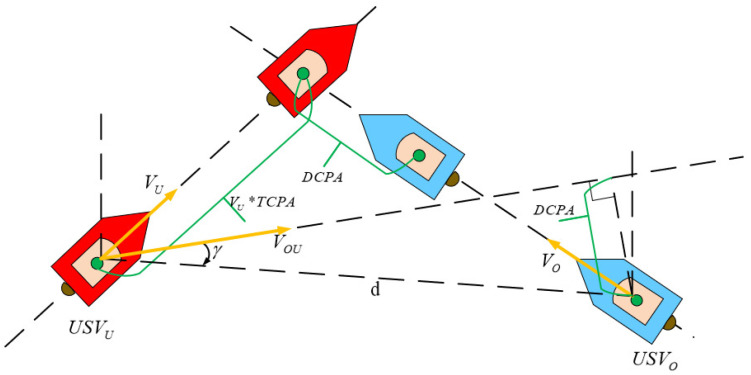
DCPA and TCPA.

**Figure 6 sensors-22-02099-f006:**
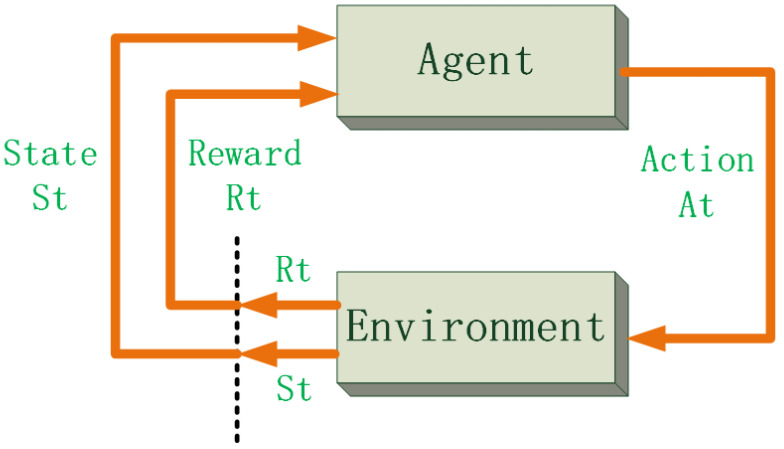
Markov decision process.

**Figure 7 sensors-22-02099-f007:**
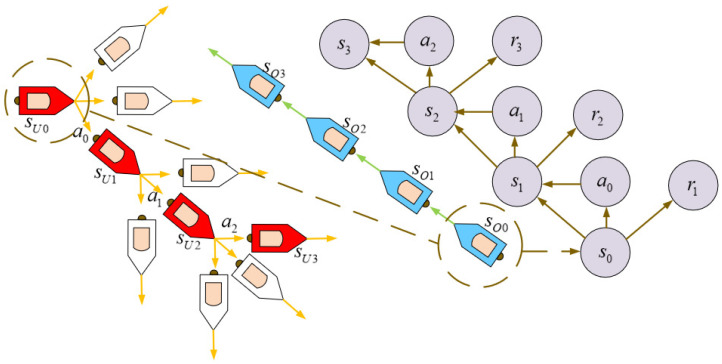
The update process of the USV collision avoidance algorithm.

**Figure 8 sensors-22-02099-f008:**
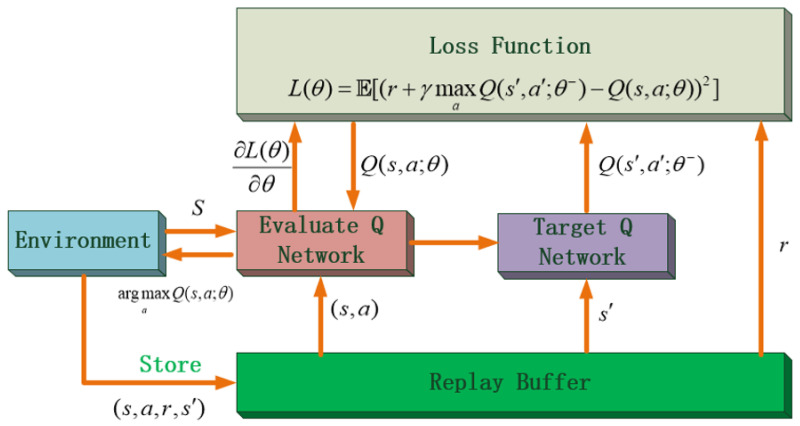
The update process of the deep Q learning algorithm.

**Figure 9 sensors-22-02099-f009:**
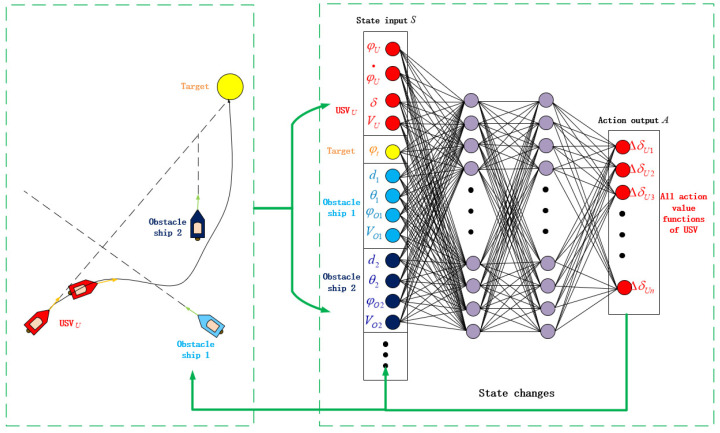
The training process of the USV agent with the reinforcement learning algorithm.

**Figure 10 sensors-22-02099-f010:**
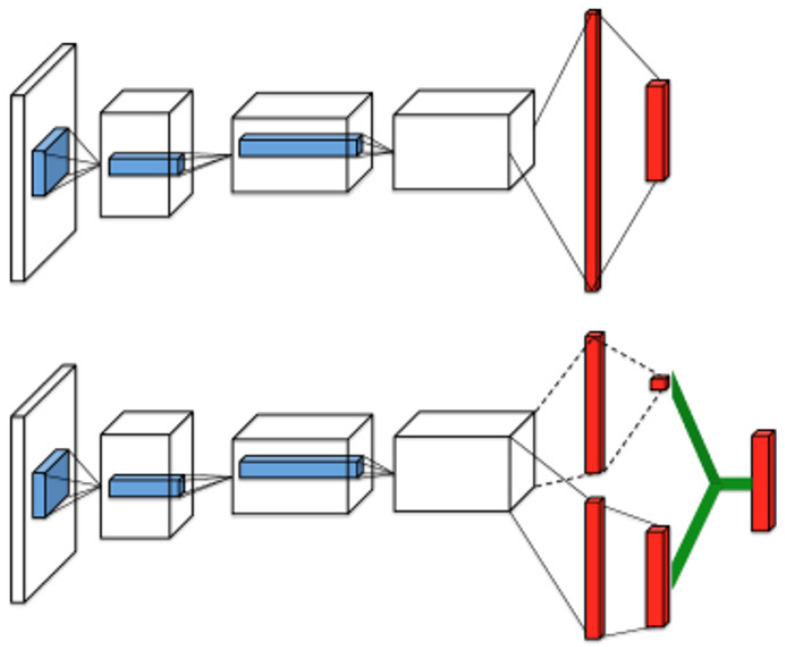
Dueling network architecture for the deep Q network.

**Figure 11 sensors-22-02099-f011:**
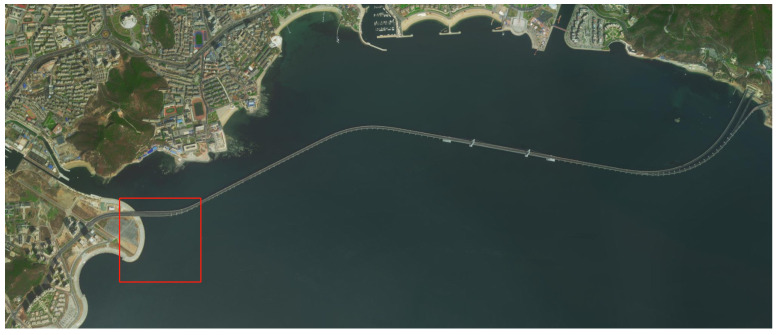
Sea area near Dalian Maritime University.

**Figure 12 sensors-22-02099-f012:**
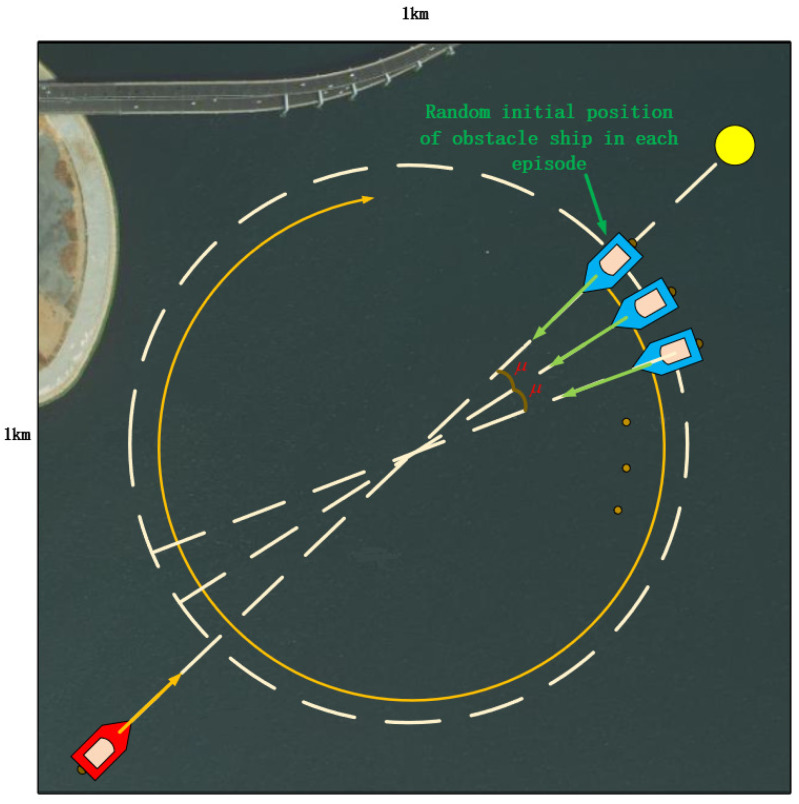
Training environment of the USV collision avoidance algorithm.

**Figure 13 sensors-22-02099-f013:**
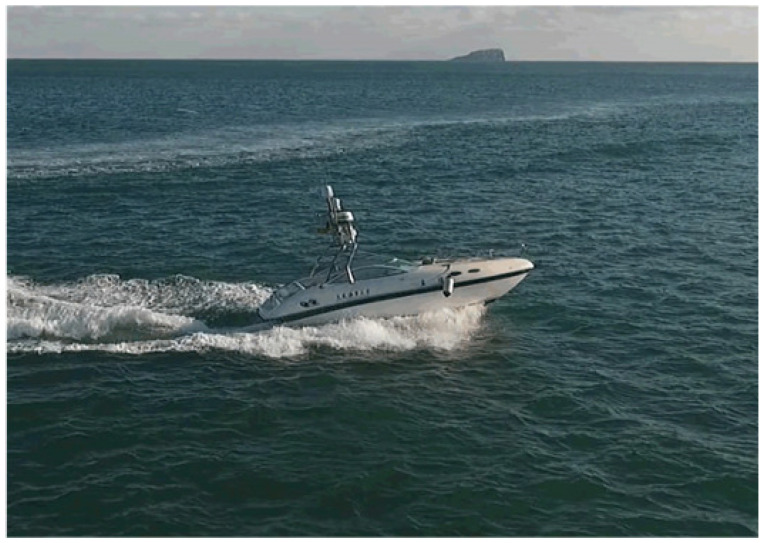
“Lan Xin” USV.

**Figure 14 sensors-22-02099-f014:**
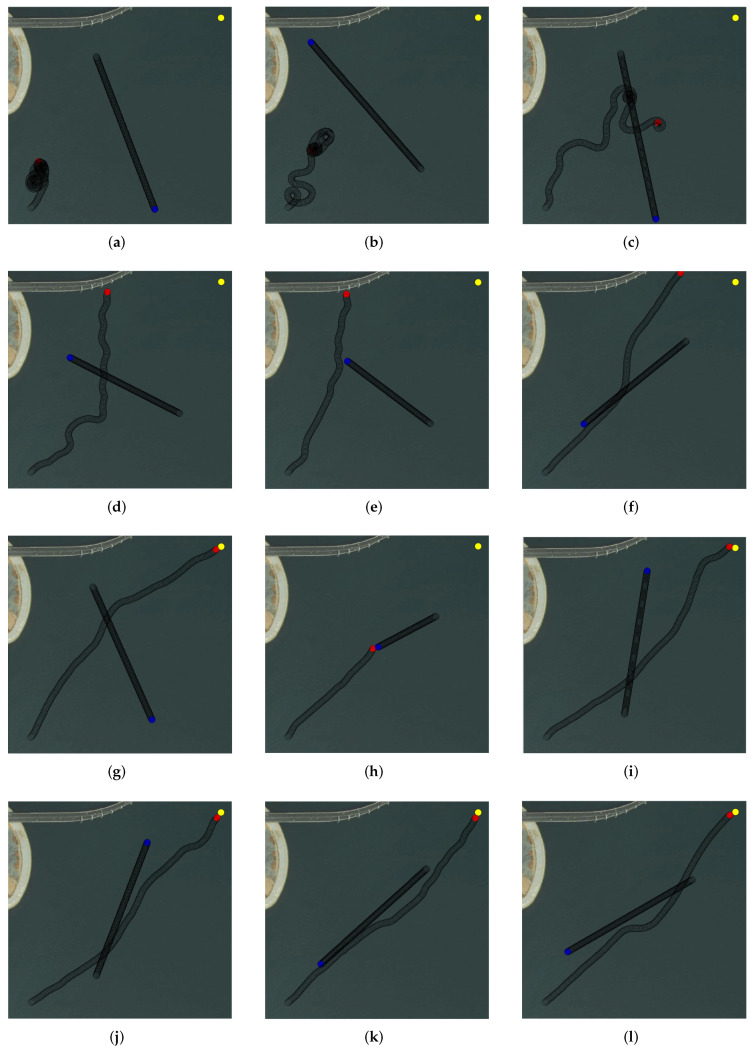
The training effect of different training stages: (**a**) 15 episodes; (**b**) 50 episodes; (**c**) 80 episodes; (**d**) 100 episodes; (**e**) 200 episodes; (**f**) 400 episodes; (**g**) 500 episodes; (**h**) 520 episodes; (**i**) 700 episodes; (**j**) 1000 episodes; (**k**) 1800 episodes; (**l**) 2000 episodes.

**Figure 15 sensors-22-02099-f015:**
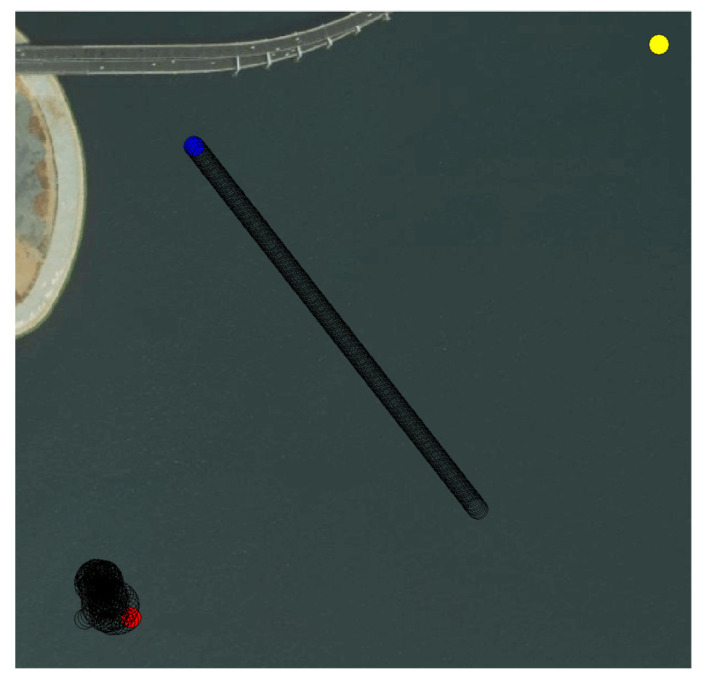
The effect of removing stage rewards.

**Figure 16 sensors-22-02099-f016:**
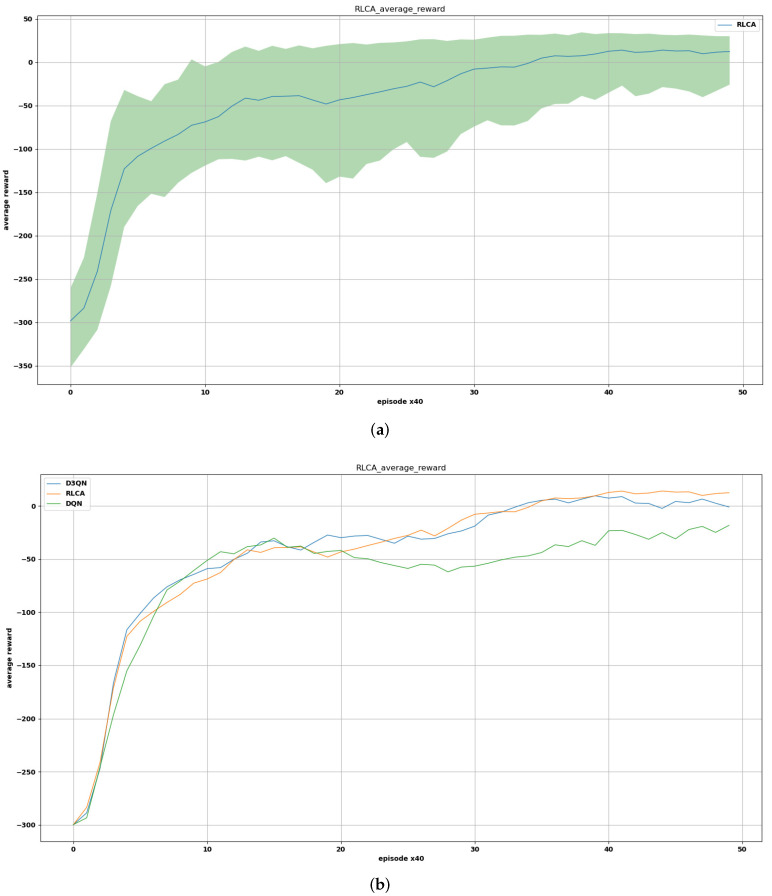
The average reward of the training process. (**a**) The average reward of the RLCA algorithm with the confidence region. (**b**) The average reward of the RLCA algorithm, D3QN algorithm, and DQN algorithm.

**Figure 17 sensors-22-02099-f017:**
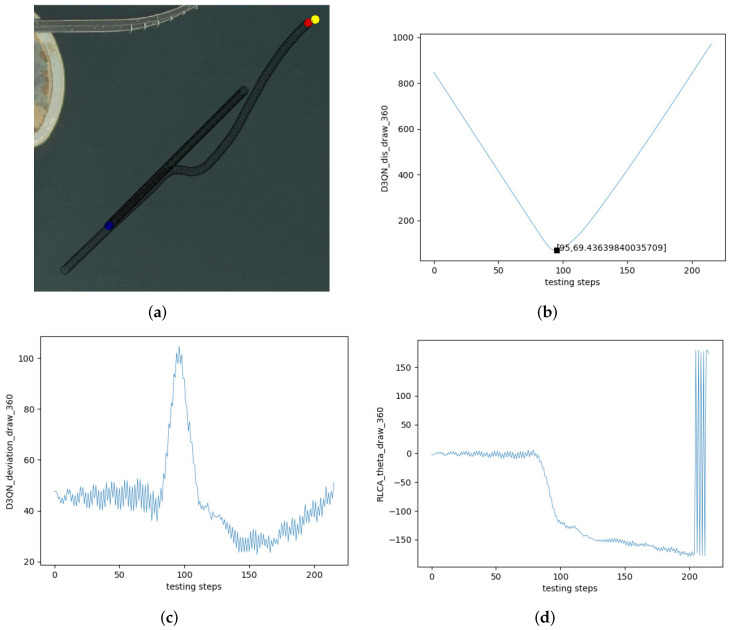
Test effect of the head-on encounter. (**a**) Head-on. (**b**) The distance between two USVs. (**c**) The course of the own USV. (**d**) θ of the own USV.

**Figure 18 sensors-22-02099-f018:**
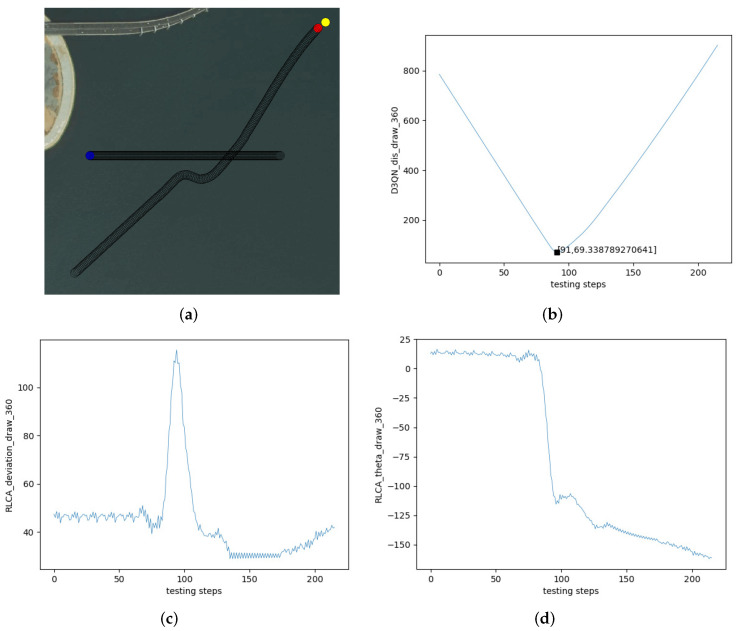
Test effect of Cross Encounter 1. (**a**) Crossing. (**b**) The distance between two USVs. (**c**) The course of the own USV. (**d**) θ of the own USV.

**Figure 19 sensors-22-02099-f019:**
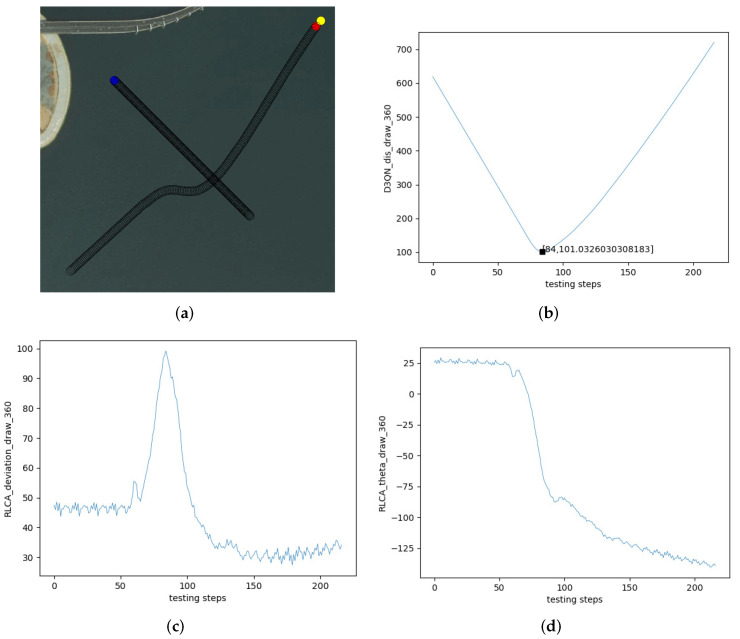
Test effect of Cross Encounter 2. (**a**) Crossing. (**b**) The distance between two USVs. (**c**) The course of the own USV. (**d**) θ of the own USV.

**Figure 20 sensors-22-02099-f020:**
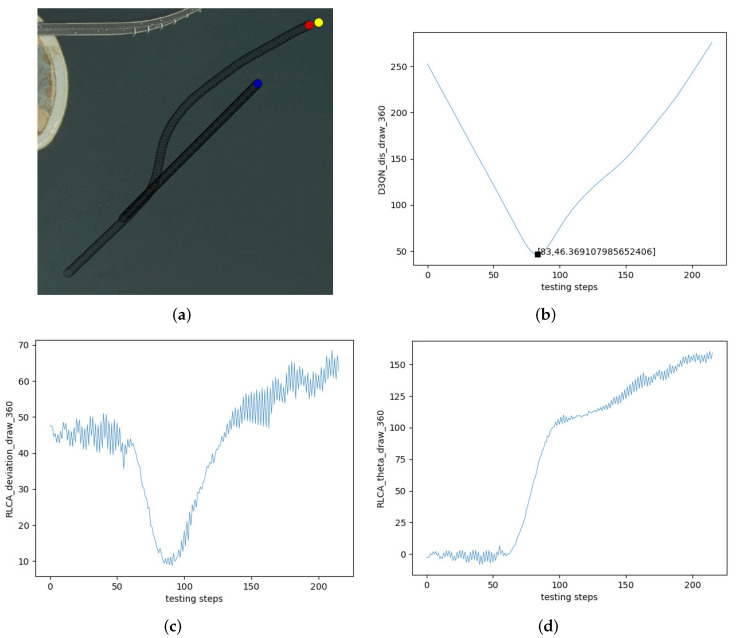
Test effect of the overtaking encounter. (**a**) Overtaking. (**b**) The distance between two USVs. (**c**) The course of the own USV. (**d**) θ of the own USV.

**Figure 21 sensors-22-02099-f021:**
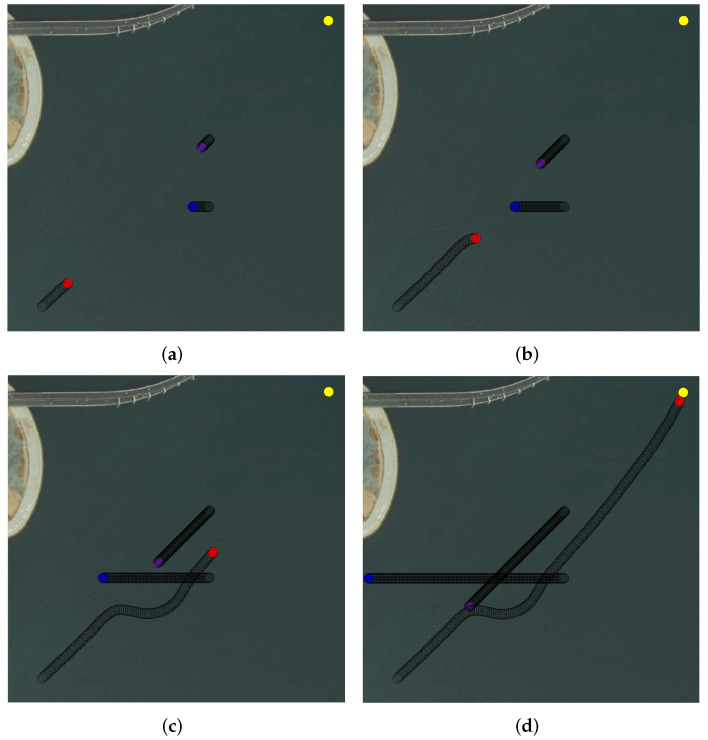
Multi-USV collision avoidance Simulation Environment 1. (**a**) Stage 1. (**b**) Stage 2. (**c**) Stage 3. (**d**) Stage 4.

**Figure 22 sensors-22-02099-f022:**
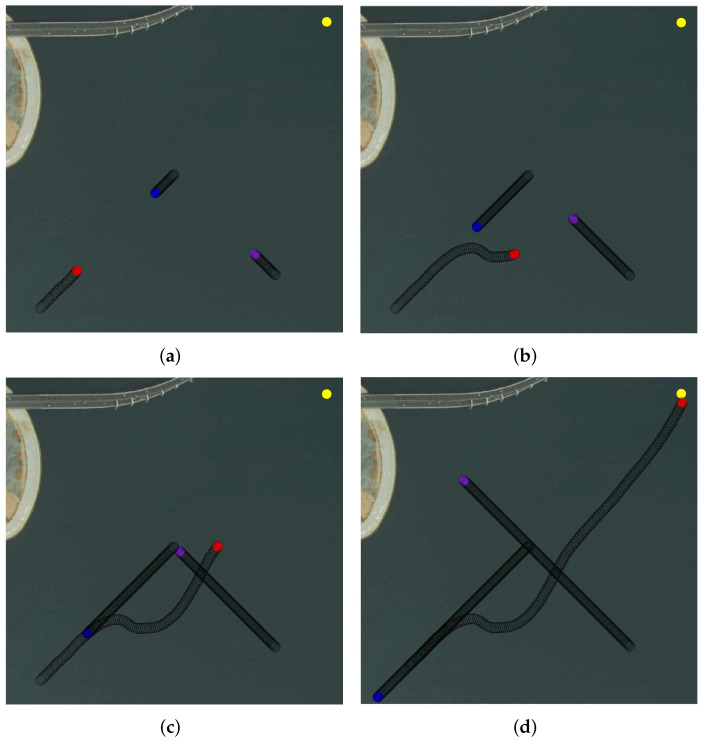
Multi-USV collision avoidance Simulation Environment 2. (**a**) Stage 1. (**b**) Stage 2. (**c**) Stage 3. (**d**) Stage 4.

**Figure 23 sensors-22-02099-f023:**
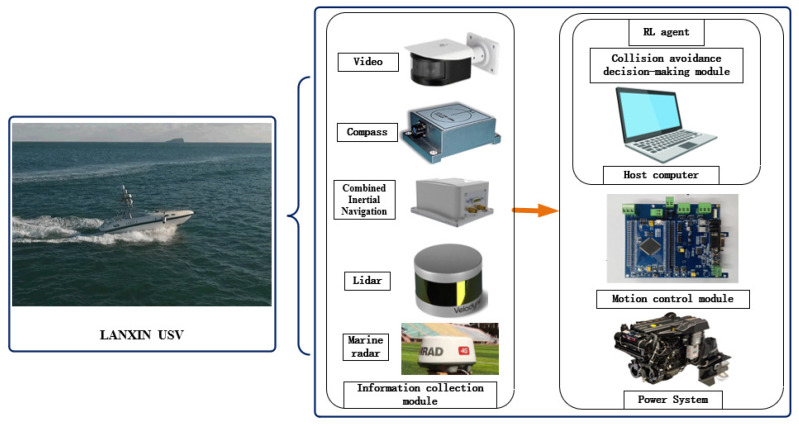
“Lan Xin” USV autonomous collision avoidance control structure.

**Table 1 sensors-22-02099-t001:** Parameters of the “Lan Xin” USV.

Parameters	Value
Length Between Perpendiculars	7.02 m
Breadth	2.60 m
Speed	≤35 kn
Draft (Full Load)	0.32 m
Block Coefficient	0.6976
Displacement (Full Load)	2.73 $m^{3}$
Rudder Area	0.2091 $m^{2}$
Distance Between Barycenter and Center	0.35 m

**Table 2 sensors-22-02099-t002:** Hyperparameters of reinforcement learning.

Hyperparameter	Value
Batch Size	600
Replay Memory Size	350,000
Target Network Update Frequency	5360
Learning Rate (α)	0.0001
Discount Factor (γ)	0.99
Initial Exploration (ε)	1
Final Exploration	0.01
Final Exploration Frame	30,000
Replay Start Size	2000

## Data Availability

Not applicable.

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
