# Peer review of "A Novel Reinforcement Learning Collision Avoidance Algorithm for USVs Based on Maneuvering Characteristics and COLREGs"

_sensors, 2022, doi:10.3390/s22062099_

Round 1

Reviewer 1 Report

The research is interesting, well structured, pertinent conclusions. 

Reviewer 2 Report

An RL-based USV collision avoidance mechanism has been proposed that incorporates a category-based exploration method into the mechanism. In Section 5, the simulation experiments have demonstrated the merits of the proposed mechanism in meeting the desired targets. Overall, the proposed method should be of significant interest to the community of researchers that are exploring similar collision avoidance mechanisms.

There are a few details in the manuscript that have not been clearly described to the reader which may lead some to misinterpret their meanings and making it more challenging to appreciate the results presented in the article. I assume that some of these details may be well-known to readers that share the same expertise in this field of research, but is not common knowledge to the general readers. They include the following:

1) In the second paragraph of Section 2.2, can you please define and explain to the reader what the acronyms DCPA and TCPA refer to in this work? This will help in better understanding the derivations in equations (1) to (8).

2) Can you further elaborate on how equations (1) and (2) are derived? I assume they rely on simple geometrical analysis, but it is quite challenging to interpret it without a clear diagram. I expected to be able to visualize it from Figure 4, but it lacks the dimensions given in equations (1) and (2). 

3) In equation (9), the parameter "r" is defined as the rate of change of the USV course. Meanwhile, in the earlier section, the same parameter is defined as the radius of the ship domain. These multiple definitions for the same parameter can cause some confusion to the reader and using the same parameter labels for different definitions should be avoided.

Reviewer 3 Report

Thank you for this extensive paper on USV collision avoidance. Overall I think the paper is very interesting. Here are some comments:

  1. In the introduction you state that COLREGs need to be followed when encountering other USVs. However, COLREGs should be followed when encountering any type of vessel. COLREGs were specifically designed for vessels commanded by humans. In fact, COLREGs may not be required at all if there were only USVs. Completely different collision avoidance behaviour could be thought of in that situation.
  2. The paragraph lines 37 to 77 is long and hard to read. I recommend grouping the content into several similarly-themed paragraphs.
  3. With my first comment in mind: how do you ensure compliance with RULE 8b? The results seem to suggest that the rule is followed. But how?
  4. Why is your ship domain and dynamic area circular? Other authors use different shapes (for instance ellipses).
  5. As shown in figures 20 and 21 your methods seems to have some difficulty with multiple vessels as the USV does a lot of steering. Could you comment on this?
